# Ultrasonography and Sonoelastography Characteristics of Benign vs. Malignant Mesenteric Lymph Nodes in Cats: An Update

**DOI:** 10.3390/ani13162664

**Published:** 2023-08-18

**Authors:** Elettra Febo, Francesca Del Signore, Nicola Bernabò, Andrea Paolini, Francesco Simeoni, Andrea De Bonis, Martina Rosto, Sara Canal, Massimo Vignoli

**Affiliations:** Department of Veterinary Medicine, University of Teramo, Località Piano D’ Accio, 64100 Teramo, Italy; elettrafebo@gmail.com (E.F.); nbernabo@unite.it (N.B.); francsimeoni@gmail.com (F.S.); adebonis@unite.it (A.D.B.); martina_rosto@hotmail.it (M.R.); scanal@unite.it (S.C.); mvignoli@unite.it (M.V.)

**Keywords:** cat, ultrasound, lymph node, sonoelastography

## Abstract

**Simple Summary:**

This study aimed to evaluate the feasibility of using a multivariate diagnostic approach by combining B-mode ultrasound, color Doppler ultrasonography, and strain elastography (SE) to distinguish benign from malignant mesenteric lymph nodes in cats. Feline enlarged mesenteric lymph nodes (LNs) were evaluated using B-mode ultrasound, color Doppler ultrasonography, and SE. Short-to-long axis ratio, borders, echogenicity, hilum, vascular flow distribution, elastographic pattern, and strain ratio were recorded. Histological and/or cytological diagnosis was available for each LN. A multivariate statistical analysis was performed including the data from B-mode ultrasound, color Doppler ultrasonography, and SE. A total of 88 LNs were included, 46 (52.3%) benign and 42 (47.7%) malignant; in the benign group, 40 LNs had a diagnosis of reactive hyperplasia (group A) and 6 eosinophilic sclerosing lymphadenitis (group B), while in the malignant group 42 had a diagnosis of lymphoma (group C). The principal component analysis approach provided evidence that by combining B-mode-based scores, the three groups of LNs can be accurately distinguished. Combining B-mode- and color Doppler-based scores with SE scores allows reactive lymph nodes to be distinguished from other lymph node abnormalities. Further studies are needed for neoplastic and inflammatory conditions in feline species.

**Abstract:**

(1) Background: Strain elastography (SE) is an ultrasound-based technique able to non-invasively assess tissue elasticity, with malignant tissues being stiffer than normal tissues. The purpose of the study was to evaluate the diagnostic performance of SE to differentiate feline mesenteric benign and malignant lymph nodes (LNs) using a multivariate approach including both SE results and B-mode ultrasound and color Doppler findings. (2) Methods: Feline enlarged mesenteric LNs were evaluated using B-mode ultrasound, color Doppler ultrasonography, and SE. Short-to-long axis ratios, borders, echogenicity, hilum, vascular flow distribution, elastographic patterns, and strain ratios were recorded. Histological and/or cytological diagnosis was available for each LN. (3) Results: A total of 88 LNs were included, 46 (52.3%) benign and 42 (47.7%) malignant; in the benign group, 40 LNs had a diagnosis of reactive hyperplasia (group A) and 6 eosinophilic sclerosing lymphadenitis (group B), while in the malignant group 42 had a diagnosis of lymphoma (group C). The principal component analysis approach showed evidence that by combining B-mode- and color Doppler-based scores with SE scores, the three groups of LNs can be accurately distinguished. (4) Conclusions: Our results demonstrate that a multivariate sonographic approach combining B-mode ultrasound, color Doppler ultrasonography, and SE can accurately distinguish benign from malignant LNs, thus helping in the clinical advice of feline patients.

## 1. Introduction

Lymph node identification and evaluation through imaging is crucial for prognosis and therapeutic options. Ultrasound (US) in particular provides the ability to visualize and evaluate lymph node characteristics in a rapid and non-invasive way, especially when assessing abnormalities in mesenteric lymph nodes. This is of paramount importance in the case of malignancies as it provides information about the actual spread of the disease and properly stages it.

However, although many B-mode ultrasound findings have been correlated to malignant infiltration (e.g., parenchymal heterogeneity, enlargement, and loss of normal hilar vascularization replaced by peripheral or mixed patterns [1,2,3,4,5]), these findings may not be specific for neoplastic infiltration and they may be recorded in inflammatory lymph nodes too [2,5,6,7,8,9,10], thus requiring cytology or histopathology for definitive diagnosis.

Elastosonography is a non-invasive imaging technique that allows for estimations of tissue stiffness through the use of a conventional sonography machine with special software and a conventional ultrasound probe [11,12]. Since pathologic tissues often exhibit altered mechanical properties, and as stiffness is often used to assess health, elastosonography can allow for differentiation between normal and diseased tissues due to differences in their elastic properties [13].

Elastosonography is based on the principle that the application of a stress force on a tissue will induce internal displacements intrinsically related to its elastic properties [14]. In strain elastography (SE), stress is applied through repeated manual compression of the transducer, and the amount of lesion deformation relative to the surrounding normal tissue is measured and displayed as a color map called an elastogram [15]. Elastograms are usually displayed as a semi-transparent overlay onto gray-scale images using a color-coded scale that varies between US systems [16]. Malignant tissues tend to be stiffer than normal tissues as they contain an increased density of tumor cells, vasculature, and fibrotic material [17]. 

Furthermore, with SE it is possible to obtain a semi-quantitative evaluation of the target tissue by comparing the relative grade of compression, expressed as elasticity index (EI), with a reference tissue. A strain ratio (SR) > 1 indicates that the target lesion compresses less than the normal reference tissue, indicating lower strain and greater stiffness [18]. 

Palpation with the assessment of tissue strain has long been a mandatory component of every medical examination. However, in the era of evidence-based medicine, this largely subjective examination has gradually begun to lose importance [19]. For these reasons, elastosonography is very popular in human cancer research and has been applied to several organs, such as the breast [20], prostate [21], thyroid [22], and lymph nodes [23]. Compared with manual palpation, elastosonography has the advantage of evaluating deeper-lying lesions and is semi-quantifiable [24]. Conventional B-mode ultrasound is often used to detect enlarged lymph nodes, and cytology or histopathology is then typically required to differentiate benign from malignant lymph nodes. Ultrasonographic findings such as size, shape, echogenicity, margination, echotexture, and vascularity are used to differentiate normal from abnormal lymph nodes [1,2,9,10,25,26,27], and some imaging criteria have been suggested as predictors of malignancy. For example, ratios comparing their short (S) and long (L) axes increase more significantly with neoplastic infiltration [2,28]. S/L axis ratios >0.7 are usually predictive of neoplasia, as opposed to ratios < 0.7, which more consistently indicate normal or reactive lymph nodes [2,4,17]. Furthermore, rounded and hypoechoic lymph nodes tend to be more malignant [4]. However, none of these criteria can be currently used to predict malignancy with certainty. Elastosonography provides new evaluation parameters for lymph node assessment compared to conventional B-mode sonography. In human medicine, it has been shown that elastosonography is a promising tool for the differentiation of benign and malignant lymphadenopathy. Increased lymph node stiffness was seen in people with metastatic squamous cell carcinoma and melanoma [29,30], and elastosonographic scores were reported to be significantly different in human benign and malignant cervical lymph nodes [16,31,32,33]. In veterinary medicine, the interest in possible elastographic applications is increasing, with several reports already available about the feasibility of its use in assessing abdominal and muscle–skeletal disorders [34,35,36,37].

As far as the specific focus on lymph nodes, SE has been investigated as a possible tool to differentiate malignant from benign lymph nodes in dogs, with or without the association of Doppler ultrasonography or contrast-enhanced ultrasonography (CEUS) findings, and this approach has shown promising results [7,17,38,39]. 

In cats, only a single report is available regarding the application of SE on lymph nodes, which used a small sample and showed overlaps between benign and malignant lymph nodes [38].

Considering that sonographic and sonoelastographic findings may not provide consistent results to differentiate benign from malignant lymph nodes, combining these two techniques may provide a more accurate diagnosis.

There are two reports available about this topic for the canine species.

The first one combined Doppler ultrasonography with SE findings to detect lymph node malignancy in heterogenous groups of dogs, including reactive, inflammatory, and neoplastic superficial lymph nodes; the second combined B-mode sonography, Doppler ultrasonography, contrast-enhanced ultrasonography (CEUS), and SE to discriminate between metastatic and non-metastatic lymph nodes in bitches with mammary carcinoma, with both of the reports describing the higher diagnostic performance of combining the techniques rather than applying a single one [7,39].

Since no similar reports applied to feline medicine are available, the purpose of this study was to evaluate the usefulness of SE in the differentiation of benign and malignant enlarged mesenteric lymph nodes in cats, providing a multivariate approach including both SE results and B-mode and color Doppler findings.

## 2. Materials and Methods

Cats with enlarged mesenteric lymph nodes detected during abdominal ultrasound at the Veterinary Teaching Hospital of the University of Teramo (from September 2020 to December 2022) were prospectively included in the study.

All the lymph nodes were evaluated ultrasonographically by a board-certified veterinary radiologist (M.V.) using B-mode sonography and color Doppler ultrasonography first and then SE with the same ultrasound system (LOGIC S8 XD clear, Ge) and an 11 MHz linear probe. In B-mode sonography, maximum short-axis (SA) diameter and maximum long-axis (LA) diameter were measured and the short-to-long axis (S/L) ratio was calculated; borders (regular or irregular), parenchymal uniformity (homogeneous or heterogeneous), hilum (present or absent), and vascular flow distribution as shown by the color Doppler mode (absent/hilar or peripheral/mixed) were also recorded.

Doppler gain was adjusted to the highest value without the presence of noise, ensuring high sensitivity for the detection of smaller and low-velocity flow vessels. Pulse repetition frequency was kept to the lowest value without aliasing artifacts.

For analyzing the diagnostic performance of B-mode sonography and color Doppler ultrasonography, scores were assigned for five criteria: S/L ratio (ratio ≤ 0.7, 1; score > 0.7, 2), borders (regular, 1; irregular, 2), echogenicity (homogenous, 1; heterogeneous, 2), hilum (present, 1; absent, 2), and vascular flow distribution as shown by the color Doppler mode (absent or hilar, 1; peripheral or mixed, 2). The US score for each LN was determined by adding the individual scores obtained by B-mode sonography and color Doppler ultrasonography. 

SE was performed immediately before histology or fine-needle aspiration (FNA) for cytology. Compression with light pressure followed by decompression was applied repeatedly by the transducer perpendicular to the lymph node. Real-time B-mode and elastographic images simultaneously appeared side-by-side on the screen. The elastographic images were displayed using a color mapping according to the degree of strain, with the color scale ranging from red (soft tissue) to blue (stiff tissue). Images were saved when the compression scale became green, indicating that the movements performed on the lesion were uniform and regular. The top of the region of the enlarged lymph node on elastography images was set to include the abdominal wall. 

Elastographic images were qualitatively and semi-quantitatively evaluated.

The qualitative analysis was performed by observing all the histograms and classifying them into 5 elastographic patterns (EPs), as described as follows based on the current literature [29] (Figure 1).

The semi-quantitative analysis was performed by measuring the EI of the lymph node and the near-field abdominal wall considered as a reference tissue [33].

EI was measured by manually tracing an area including as much parenchyma as possible in each lymph node and the near-field reference tissue. EI values ranged from 0 to 6; a higher value indicated greater stiffness and a color closer to blue on the elastogram. 

The semi-quantitative SR evaluations were computed between the two EI values through the use of the E-RATIO FUNCTION (GE Healthcare). An SR value of >1 represented increased tissue stiffness in the skin nodule relative to the reference healthy tissue selected.

Cytological and/or histological diagnosis was available for each lymph node. Decisions for assigning lymph nodes to benign or malignant groups were based on the criteria described below.

Reactive lymph nodes had predominantly small lymphocytes with increased intermediate and large lymphocytes and scattered plasma cells; reactive lymph nodes could also have scattered neutrophils, occasional macrophages and eosinophils, and rare mast cells. 

Lymph nodes were classified as inflammatory if they had more than 5% neutrophils, more than 3% eosinophils, or frequent macrophages for granulomatous inflammation. Lymph nodes were defined as malignant if there were >50% lymphoblasts (lymphoma) and if there was a neoplastic population of epithelial, mesenchymal, or round cells (metastatic neoplasia). US-guided sampling was performed on each lymph node.

Histopathology was performed in the case of nondiagnostic samples when the owners agreed to surgical excision or post-mortem excision.

The excised lymph nodes were then preserved in 10% formalin and submitted for histopathological analysis. The lymph nodes were examined at multiple levels of the paraffin block and hematoxylin and eosin staining was used. 

Lymph nodes with doubtful cytological diagnoses not confirmed by histopathology were not included. 

Statistical analysis was performed with GraphPad Prism v9.2.0.332.

The nonnormal distribution of quantitative variables was determined by the Kolmogorov–Smirnov test, with values reported as median and interquartile range.

Quantitative variables, including US score, EP, and SR, were compared between neoplastic and nonneoplastic nodes using a Kruskal–Wallis test, and analyses were considered significant at *p* < 0.05. 

US mode score, EP, and SR were used to provide a multivariate statistical analysis through principal component analysis (PCA)

## 3. Results

A total of 34 cats and 88 LNs were included with a median age of 6.5 years (1–16); 35.3% (*n* = 12) were spayed females and 69.7% (*n* = 22) were castrated males. 

Of the 88 LNs examined, 46 (52.3%) were benign and 42 (47.7%) were malignant based on histological and/or cytological diagnosis; of the 46 benign LNs, 40 had been reported as reactive hyperplasia and 6 as eosinophilic sclerosing lymphadenitis. The 42 malignant LNs included 38 LNs classified as high-grade lymphoma and 4 LNs categorized as large granular lymphoma (LGL) (Table 1). No clonality test was performed for lymphoma.

The lymph nodes were divided into three groups: group A for reactive hyperplasia, group B for eosinophilic sclerosing lymphadenitis, and group C for lymphoma. The median US score for group A was 5.6 (5–8), the median score for group B was 9.5 (9–10), and the median score for group C was 8.4 (7–9); a statistically significant difference was observed between all three groups (*p* = 0.01).

Regarding the SE results, in group A EP 1 was recorded in 36% of lymph nodes (*n* = 15/40) (Figure 2), EP 2 was recorded in 35% of cases (*n* = 14/40), and EP 3 was recorded in 29% of cases (*n* = 11/40). In group B, EP 2 was recorded in 100% of cases (*n* = 6/6) (Figure 3), while in group C EP 2 was recorded in 4% of cases (*n* = 2/42), EP 3 was recorded in 36% of cases (*n* = 15/42), EP 4 was recorded in 58% of cases (*n* = 24/42) (Figure 4), and EP 5 was recorded in 2% of cases (1/42%).

The median EP was 2 (1–3) for group A, 5 (5–5) for group B, and 4 (2–5) for group C; a statistically significant difference was observed between all three groups (*p* = 0.01).

The median SR was 0.5 (0.1–0.9) for group A, 1.8 (1.5–2.2) for group B, and 0.7 (0.1–1.6) for group C.

All of the results are presented in Table 2.

The results of PCA are presented in Figure 5.

The graph shows the clear division of the cases included in the three groups, with all the hyperplastic lymph nodes divided from the fibrotic and neoplastic lymph nodes, with slight overlaps between the three groups. 

The elements labeled with A are the lymph nodes of group A (reactive lymph nodes), the elements labeled with B are the lymph nodes of group B (eosinophilic sclerosing lymph adenitis), and the elements labeled with C are the lymph nodes of group C (lymphoma). The X and Y axes are coordinates aimed to help order the data in space and assess the degree of variability between the elements. This graph aims to represent how much the combined data from US and SE are able to classify the lymph nodes in the three groups. It is obvious that reactive lymph nodes do not overlap with those with lymphoma, except for a few elements highlighted with blue circles; the elements of group B are indicated by red circles, and it can be observed that there is only a slight overlap with group C.

## 4. Discussion

This work aimed to evaluate the association between B-Mode- and color Doppler-based scores and sonoelastographic findings to distinguish benign from malignant mesenteric lymph nodes in feline species.

Cats are often referred to veterinary facilities to perform assessment of abdominal US chronic gastrointestinal symptoms. Despite US being an essential step for diagnostic workup, sometimes it cannot discriminate a lymphoplasmacytic enteritis from low-grade lymphoma [40]. This is also applied to mesenteric lymph nodes, where the distinction between inflammatory and neoplastic lymph nodes may not be immediately obvious when assessed by US, thus requiring the need for cytology or histopathology for definitive diagnosis [41,42].

We hypothesized that providing a multivariate approach including both US and sonoelastography could improve diagnostic accuracy and help to differentiate benign from malignant lymph nodes. Furthermore, the exclusive distinction between benign and malignant lymph nodes was supposed to be reductive based on the evidence that a high fibrotic component in inflammatory lymph nodes could be an important bias for sonoelastographic evaluation; for that purpose, we decided to divide the lymph nodes included into three groups: reactive, fibrotic, and neoplastic lymph nodes.

As for the sonographic evaluation, borders, echogenicity, the presence of hilum, patterns of vascularization, and S/L ratio were recorded for each lymph node, thus providing a scoring rubric as previously described in the text. This choice was based on reports in the current literature that describe the single feature as not being accurate enough to distinguish benign from malignant lymph nodes, and the scores based on the combined findings provided more accurate diagnoses, as described in the recent literature available for canine species and people [7,43].

The contour regularity, parenchymal uniformity, and nodal hilum definition can overlap between benign and malignant lymph nodes [4,39]. In particular, uniform enlargement is frequently described in the case of lymphoma, especially for the canine species [2,6,7]; even if parenchymal heterogeneity is frequently observed in malignant nodes because of focal areas of calcification, hypoechoic areas, and focal nodular lesions, the presence of homogeneous parenchyma cannot rule out the presence of metastases [5,39,44,45].

Moreover, the absence of hilum due to metastatic dissemination can be observed in the relatively late stage of the disorder, thus making it an unreliable finding in early metastatic dissemination [7].

According to some authors, evaluation of lymph node vascularization using the Doppler technique offers a very good appreciation of their status, suggesting the possibility of differentiating metastatic from benign or unaffected lymph nodes, with hilar/absent vascularization indicative of normal lymph nodes and peripheral and hilar blood flow potentially indicative of malignancy [1,4].

However, in metastatic nodes, the ineffectiveness of color Doppler ultrasonography is described, where due to the presence of necrotic areas only hilar vascularization is found [6,7].

Finally, the size of a lymph node, as established by measuring the two axes for the differentiations of metastatic lymph nodes, is not a relevant criterion alone to distinguish benign from malignant lymph nodes due to the high variability in shape, especially for round ones [4,6,7,46].

If we focus on every single sonographic finding from the lymph nodes described herein, our results are consistent with the previous literature. Indeed, none of the findings appear to be specific to benign or malignant lymph nodes, especially in the case of eosinophilic sclerosing lymphadenitis, which shares many of the features observed in the case of lymphoma. Comparing, on the contrary, the scores provided by the sums of specific findings, reactive lymph nodes express a significant difference compared to fibrotic or neoplastic lymph nodes, with a median score < 6. This result is consistent with the available human literature, which classifies as benign the lymph nodes characterized by a low score [47]. The interesting aspect observed herein is that in the case of sclerosing eosinophilic lymph adenitis, the score is higher than that for both reactive and lymphomatous lymph nodes; this particular condition, indeed, is often misdiagnosed as neoplasia with US alone due to the presence of intestinal masses and severe lymph adenomegaly with heterogeneous parenchyma, thus requiring cytology or histopathology for definitive diagnosis [48,49,50].

Regarding the application of SE, the available reports describe that this technique alone may not provide enough specific information to classify the lymph nodes; for both dogs and cats, there is an overlap in terms of elastographic appearance between benign and malignant lymph nodes [38,39]. Despite the understanding that lymph nodes are supposed to be stiffer than benign ones, malignant lymph nodes may be characterized by a relatively soft appearance if there is necrotic core tissue, as happens in the case of lymphoma [33,38,51].

The results described herein are consistent with the previous data: reactive lymph nodes appear soft with significantly lower EP and SR values than fibrotic and lymphomatous nodes. The fibrotic lymph nodes were characterized, as expected, by the highest stiffness recordable, while in the case of lymphoma, the most consistent pattern number was four, characterized by peripheric stiff tissue and central soft parenchyma consistent with tissue necrosis.

PCA analysis based on scores, EPs, and SRs confirmed and highlighted that the three groups are characterized by different features and that they are distinct; even if PCA provides only a qualitative evaluation, the graph clearly shows evidence that the overlaps between the groups are slight, especially for reactive lymph nodes.

The results described herein evidence that the strict distinction between benign and malignant lymph nodes cannot be achieved by combining US scores and SE results since eosinophilic sclerosing lymphadenitis may express a high US score and stiff parenchyma, as in the case of lymphoma. The data reported herein evidence that sclerosing lymph nodes are even stiffer than lymphoma, but the low numerosity of this sample asks for more cases to confirm these data. Based on these results, however, lymph nodes can be safely classified as reactive if they present a US score lower than six and a soft elastographic pattern with an SR < 1. In contrast, in the case of a high US score and stiff parenchyma, further investigations are needed to exclude neoplasia. These results, even if encouraging, are not without limitations related to both the SE procedure and the results described herein.

SE limitations include reliance on the need for manual pressure in association with the probe to be performed, thus increasing the operator’s influence on the results obtained. Since the manual pressure is not directly quantifiable, the numeric measurements obtained from the procedure are considered semi-quantitative.

Other elastographic techniques, such as shear wave elastography, do not require manual pressure to obtain the elastogram, and the stiffness measurements are quantitative and provide a more objective result than SE. Even if SE provides a visual indicator of the proper pressure to apply to obtain reliable information, it is still more operator-dependent and requires proper training for the operator to be able to obtain the most valuable possible results in clinical practice [52]. 

Regarding the data, the first main limitation is that lymph node groups were not of a similar size since the fibrotic group was much smaller than the reactive and neoplastic ones.

This group, in particular, was separated from other benign lymph nodes during the process of the work since the authors were concerned about bias due to the high rate of fibrosis detectable with elastography decreasing the accuracy of SE. Indeed, the lymph nodes, even if not neoplastic, expressed the highest stiffness detectable for the presence of highly fibrotic tissue. Furthermore, only reactive lymph nodes were included in group A and the authors could not include other inflammatory lymph nodes, thus making it difficult to translate the evidence reported herein to other lymph adenitis. Another limitation relies on the inclusion of exclusively lymphomatous lymph nodes; including other neoplastic disorders (such as carcinoma or mast cell tumors) in future studies could improve our knowledge about possible specific sonographic or sonoelastographic futures. The last limitation relies on the fact that histopathology was not available for all the lymph nodes included, thus limiting a more extensive tissue diagnostic evaluation. Achieving the final diagnosis of lymphoma may indeed be highly challenging, especially for feline species. Even though high-grade lymphoma and LGL can be easily diagnosed with cytology, especially LGL which is characterized by characteristic cytoplasmic granules [53], it may not be possible to diagnose low-grade lymphoma through the sole use of cytology because of a higher prevalence of small cell types that make it difficult to distinguish between normal lymphocytes and reactive ones [54]. In the case included herein, no low-grade lymphomas were available, and it could be interesting in future studies to compare them to reactive lymph nodes. Even if the authors excluded doubtful cytology results and performed, where needed, histopathology to confirm reactive hyperplasia, they are fully aware that with cytology alone it is possible to misdiagnose low-grade lymphoma for reactive hyperplasia, and the clinical history of the patient is thus crucial to guiding clinicians toward choosing the proper diagnostic tests and taking into proper consideration all the information available from ultrasound results.

## 5. Conclusions

A multivariate diagnostic approach including sonographic and sonealstographic findings can differentiate reactive lymph nodes from eosinophilic sclerosing lymph adenitis or lymphoma, thus potentially reducing the need for further tissue sampling. A low US score and an EP of 1–3 are highly indicative of reactive lymph nodes compared to fibrotic and neoplastic ones. 

The high scores detected for the latter two groups in terms of US scores and SE require a larger number of subjects to be included to fully understand if these techniques could help in the clinical management of feline patients. 

## Figures and Tables

**Figure 1 animals-13-02664-f001:**
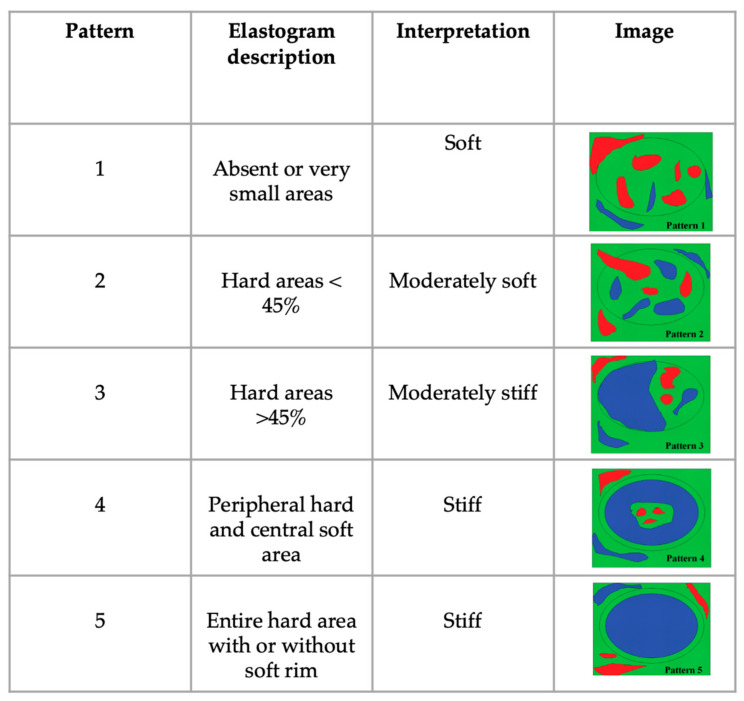
Elastographic patterns as determined by the distribution and percentage of hard areas (colored blue) in the lymph node.

**Figure 2 animals-13-02664-f002:**
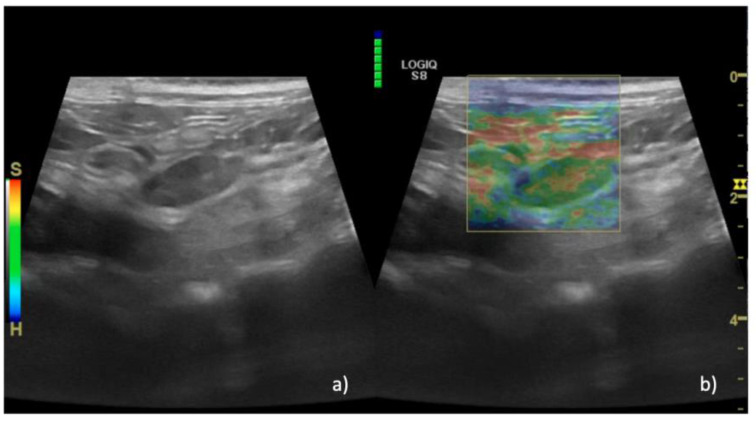
B-mode and elastographic images of a lymph node with reactive hyperplasia that show EP 1 (absent or very small hard areas). In panel (**a**), the B-mode image of the lymph node is shown, while in panel (**b**) the superimposition with the color map is shown, which is almost absent of blue areas, meaning that the parenchyma is soft. The green scale on the top left of panel b indicates the correct pressure of the probe.

**Figure 3 animals-13-02664-f003:**
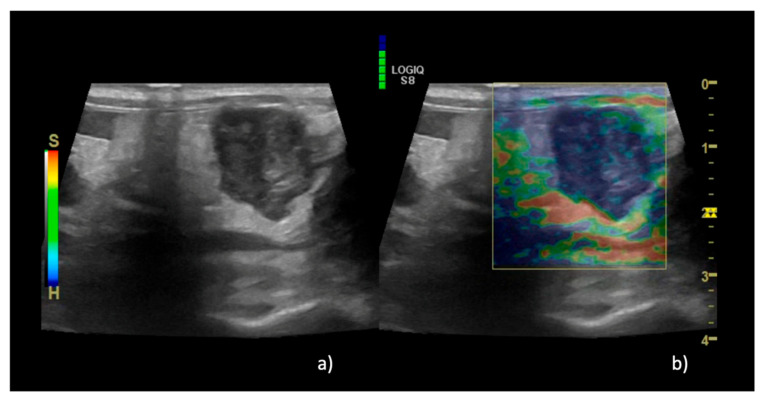
B-mode and elastographic images of a lymph node with eosinophilic sclerosing lymphadenitis that show EP 5 (entire hard area with or without a soft rim). In panel (**a**), the B-mode image of the lymph node is shown, while in panel (**b**) the superimposition with the color map is shown, with almost the entire area being blue. The green scale on the top left of panel b indicates the correct pressure of the probe.

**Figure 4 animals-13-02664-f004:**
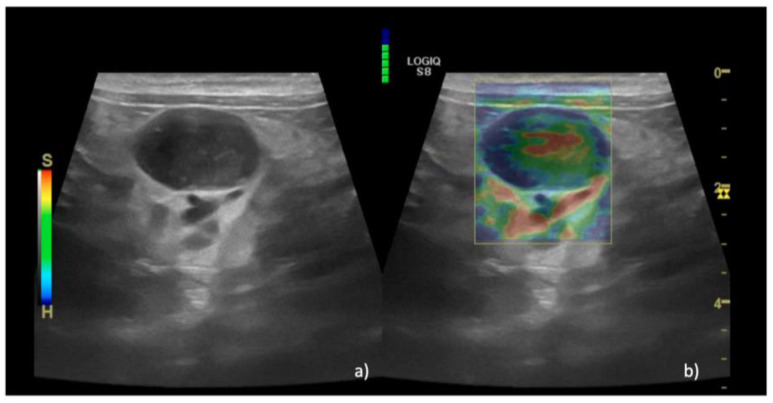
B-mode and elastographic images of a lymph node with lymphoma that show EP 4 (peripheral hard area and central soft area). In panel (**a**), the B-mode image of the lymph node is shown, while in panel (**b**) the superimposition with the color map is shown, with a peripheric blue contour and a central soft area. The green scale on the top and the left of panel b indicates the correct pressure of the probe.

**Figure 5 animals-13-02664-f005:**
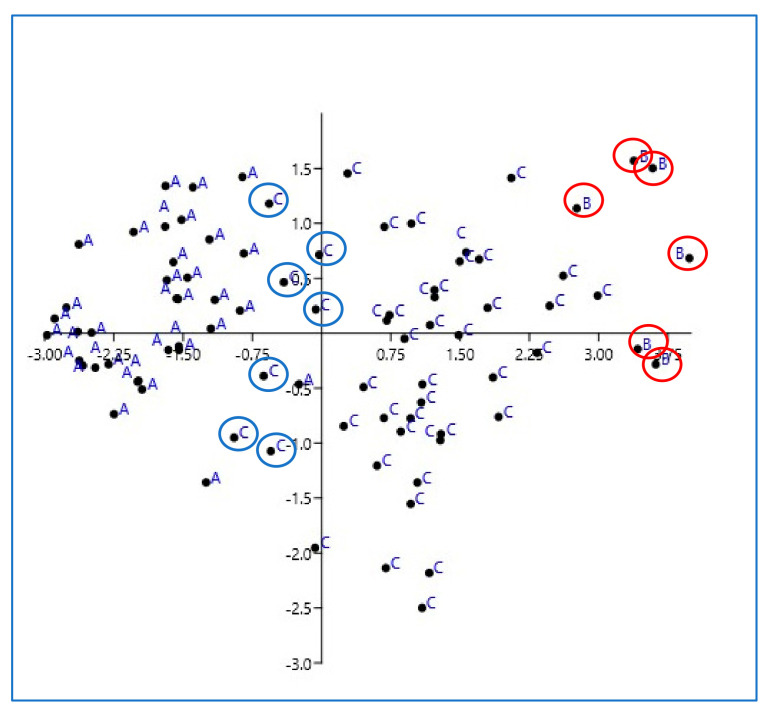
Graph of PCA results.

**Table 1 animals-13-02664-t001:** In this table, the diagnostic techniques used to achieve final diagnosis are reported.

	Cytology	Histology	Cytology/Histology
**Reactive hyperplasia**	*n* = 31 (77.5%)	*n* = 4 (10%)	*n* = 5 (12.5%)
**Eosinophilic sclerosing lymphadenitis**			*n* = 6 (100%)
**Lymphoma**	*n* = 26 (62%)	*n* = 16 (38%)	

**Table 2 animals-13-02664-t002:** US score and SE findings for each group. Data are expressed as median and interquartile range (IQR).

	Reactive Lymph Nodes Group A	Eosinophilic Sclerosing Lymphadenitis Group B	Lymphoma Group C	*p*-Value
B-mode and color Doppler				
**Borders**				
*regular*	*n* = 29 (77%)	*n* = 0 (0%)	*n* = 21 (50%)	
*irregular*	*n* = 11 (23%)	*n* = 6 (100%)	*n* = 21 (50%)	
**Echogenicity**				
*homogeneous*	*n* = 12 (28.5%)	*n* = 0 (0%)	*n* = 4 (11%)	
*heterogeneous*	*n* = 26 (71.5%)	*n* = 6 (100%)	*n* = 38 (89%)	
**Hilum**				
*present*	*n* = 37 (98%)	*n* = 0 (0%)	*n* = 0 (0%)	
*absent*	*n* = 3 (2%)	*n* = 6 (100%)	*n* = 42 (100%)	
**Vascularization**				
*hilar/absent*	*n* = 31 (82.85%)	*n* = 0 (0%)	*n* = 2 (4%)	
*peripherical/mixed*	*n* = 9 (17.15%)	*n* = 6 (100%)	*n* = 40 (96%)	
**S/L axis**				
*<0.7*	*n* = 40 (100%)	*n* = 3 (50%)	*n* = 22 (54.5%)	
*>0.7*	*n* = 0 (0%)	*n* = 3 (50%)	*n* = 20 (45.5%)	
**US score,** **Median (IQR)**	5.6 (5–8)	9.5 (9–10)	8.4 (7–9)	0.01
**Sonoelastography**				
**EP**	1 *n* = 15 (36%)	1 *n* = 0	1 *n* = 0	
	2 *n* = 14 (35%)	2 *n* = 0	2 *n* = 2 (4%)	
	3 *n* = 11 (29%)	3 *n* = 0	3 *n* = 15 (36%)	
	4 *n* = 0	4 *n* = 0	4 *n* = 24 (58%)	
	5 *n* = 0	5 *n* = 6 (100%)	5 *n* = 1 (2%)	
**Median (IQR)**	2 (1–3)	5 (5–5)	4 (2–5)	0.01
**SR, Median (IQR)**	0.5 (0.1–0.9)	1.8 (1.5-2.2)	0.7 (0.1–1.6)	0.01

## Data Availability

All the data are provided in the manuscript.

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
