# Peer review of "Ultrasonography and Sonoelastography Characteristics of Benign vs. Malignant Mesenteric Lymph Nodes in Cats: An Update"

_animals, 2023, doi:10.3390/ani13162664_

Round 1
Reviewer 1 Report
The introduction provides sufficient background and include all relevant references. The research design is appropriate and the methods adequately described. The inclusion of 6 cases of eosinophilic sclerosing lymphadenitis is really interesting. The combination of B mode, Doppler and SE could provide a helpful score in clinical evaluation in cats.
1 Differentiation between Lymphoplasmacytic enteritis (LPE) and low-grade intestinal T cell lymphoma (LGITL) is really difficult even with histology or clonality tests. However high grade lymphoma is a totally different entity. Therefore authors, as they mention lymphoma, should include a paragraph explaining which type of lymphoma (presumably high grade) were included. Maybe a table which shows the method for diagnostic (cytology or biopsy) and the use or not of other test as immunohistochemistry or PARR could be useful.
2 In the other hand dealing with the fact that cytology many times overlaps reactive with low-grade lymphoma or even normal lymph node, authors should mention this limitation. Maybe some lymph node was evaluated cytologically as reactive and could be a small cell lymphoma. The sentence “The last limitation relies on the fact that not all the lymph nodes included histopathology was available, thus limiting a more extensive tissue diagnostic evaluation.” is not enough to reflect this limitation.
3 The sentence “thus potentially reducing the need for further tissue sampling” does not reflect the goal of the study. Maybe “helping in the clinical advice of the feline patient” or something similar could be more accurate. Many cases are really complex and even with biopsies or more ancillary tests the diagnosis is not clear. So, SE should serve to add more information, not to replace the use of cytology or histology.
Minor editing of English language is required. P.e. "Despite US being an essential step for diagnostic workup, sometimes US cannot the 227 discriminate against a lymphoplasmacytic enteritis from low-grade lymphoma (39). 228
Author Response
We thank you very much for your comments,
As follows you’ll find a reply point by point to your comments; the changes in the text are highlighted with the word track changes.
1 Differentiation between Lymphoplasmacytic enteritis (LPE) and low-grade intestinal T cell lymphoma (LGITL) is really difficult even with histology or clonality tests. However high grade lymphoma is a totally different entity. Therefore authors, as they mention lymphoma, should include a paragraph explaining which type of lymphoma (presumably high grade) were included. Maybe a table which shows the method for diagnostic (cytology or biopsy) and the use or not of other test as immunohistochemistry or PARR could be useful.
We improved the description of cytology and histology part in the results and discussion. Immunohistochemistry and PARR were unfortunately available.
2 In the other hand dealing with the fact that cytology many times overlaps reactive with low-grade lymphoma or even normal lymph node, authors should mention this limitation. Maybe some lymph node was evaluated cytologically as reactive and could be a small cell lymphoma. The sentence “The last limitation relies on the fact that not all the lymph nodes included histopathology was available, thus limiting a more extensive tissue diagnostic evaluation.” is not enough to reflect this limitation.
We totally agree with this point, and it has been more extensively discussed in the main text.
3 The sentence “thus potentially reducing the need for further tissue sampling” does not reflect the goal of the study. Maybe “helping in the clinical advice of the feline patient” or something similar could be more accurate. Many cases are really complex and even with biopsies or more ancillary tests the diagnosis is not clear. So, SE should serve to add more information, not to replace the use of cytology or histology
Text has been modified accordingly.

Reviewer 2 Report
see attached pdf file

see attached pdf file
Author Response
We thank you very much for your comments,
As follows you’ll find a reply point by point to your comments; the changes in the text are highlighted with the word track changes.
Line 12: The acronym LNs is not preceded by its meaning.
The complete name has been added
Line 15: The acronym LN is not preceded by its meaning.
Text has been corrected
Line 16: The acronym US is not preceded by its meaning.
Text has been corrected
Line 19: The acronym PCA is not preceded by its meaning.
Text has been corrected accordingly
Line 32: The acronym PCA is not preceded by its meaning.
Text has been corrected accordingly
Line 45: A synonym od “disease” should be used to avoid repetition of this word.
Text has been corrected accordingly
Line 51: Sonoelastography is not a new imaging technique, I suggest removing the term “new” from this sentence.
Text has been corrected accordingly
Line 73: Sonoelastography is not a new imaging technique, I suggest removing the term “novel” from this sentence.
Text has been corrected accordingly
Lines 78-85: the authors should expand the references on the use of ultrasound in the distinction between malignant and benign lymph nodes in veterinary medicine. They could cite the following paper: “Pathologic correlation of resistive and pulsatility indices in canine abdominal lymph nodes”. Prieto et al. Vet Radiol Ultrasound 2009 ; 50(5):525-9. doi: 10.1111/j.1740- 8261.2009.01580.x
The reference list has been updated
Line 98: The acronym CEUS is not preceded by its meaning.
Text has been corrected accordingly
Figure 1: to be clearer and more intuitive the Figure 1 should contain all the relevant information,
below is an example of how the table/figure could be organized:
PATTERN ELASTORGAM DESCRIPTION ELASTOGRAM IMAGE INTERPRETATION
1 Absent or very small hard area soft IMAGE OF PATTERN 1
This figure has been modified as requested.
Line 162: The authors should specify whether the cytological and/or histological samples were obtained under ultrasound guidance or not (e.g. laparotomy or removal and subsequent histological examination of the lymph node).
Text has been modified accordingly
RESULTS:
Line 175: The authors should specify how many LNs were examined by cytological examination, how many by histological examination and how many by both methods.
A table has been added to the main text
Lines178-179: The three study groups should be associated with the 3 letters A, B and C both in the text and in table 1 as reported in lines 217-220. Authors should choose whether to identify them as groups 1, 2 and 3 or as groups A, B and C.
The three groups have been labeled with A, B and C respectively and the text has been modified accordingly.
Lines 182-183, lines 184-189 and lines 208-211: These results should be tabulated for easier viewing and interpretation. Below is an example of a table you could use to represent the results of your study.
Data have been resumed in a table as suggested.
Lines 213-215: Authors should better explain how to read this graph (Figure 5). Which quantity is expressed on the Y axis and which on the X axis? Could colors be used to distinguish the 3 groups A, B and C?
The X and Y axes are spatial coordinates aimed to order in the space the data and assess the degree of separation of the points, the more the data are “mixed” in the space and less the used test is able to differentiate the data, this explanation has been added to the main text; colors have also been added in the figure to highlight the degree of superimposition of the data analyzed
What is the reference value (mean and standard deviation or median and interquartile range) given by the sum of the 3 scores in malignant lymph nodes? And which in benign lymph nodes? Is the score difference between the 3 groups statistically significant? From the results it is not clear whether the sum of the 3 scores (B-mode, EP and SR) gives a reference value that allows clinicians to distinguish malignant from benign lymph nodes. This would satisfy the objective of the study reported in lines 111-114.
In this work we did not provide a unique score from B-mode, EP and SR because it was not the exact aim of the planned work, this aspect of the work was not properly highlighted in the introduction.
The aim of the work was to provide a multivariate approach base on B-mode, Color Doppler and SE finding; for that purpose, we provided a scoring rubric from B-Mode and color doppler findings based on the current literature described in the text and, separately, the results from SE.
Then these results were combined through PCA analysis; the graph obtained evidence that only few overlaps are evident between the groups included, thus pointing that combining the US score with SE the clinician could improve the diagnostic accuracy of the single techniques.
The aim of the study has been better clarified in the text.
DISCUSSION:
The authors should report the clinical relevance of this study. Do the results of this study allow us to distinguish benign from malignant lymph nodes by combining B-mode, Doppler and SE methods?
According to the results herein presented the strict distinction between benign and malignant lymph nodes is not totally available combining B-mode, doppler and SE methods, since the inflammatory lymph nodes and the neoplastic ones shares many features, with both US and SE techniques. However, the results evidence that low US score and EP from 1-3 are highly indicative of reactive ones. This point has been clarified in the discussion and conclusion.
Lines 274-276: this result should be reported, together with the p value, also in the results section (table)
All the results have been pointed in table 2
Lines 302-304: this result should be reported, together with the p value, also in the results section (table)
All the results have been pointed in table 2
Lines 306-322: the authors should discuss the limitations of the method (Strain elastography).
The discussion has been properly updated.

Reviewer 3 Report
Nice work in general. Well described and presented. I have few questions/comments about your work:
- In Material and methods, line 124, please clarify the criteria for homogeneous/heterogeneous classification of the lymphonodes. A description of the ultrasonographic heterogenicity would be great.
- Same section, line 125 and 129, the presence of the Doppler flow can be easily modified with the equipment settings (like PRF's, wall filters, doppler gain, etc...). How can you ensure the total absence of doppler signals? Please clarify the settings you used during the Doppler imaging.
- For the histological analysis, why they are not all histologies but some cytologies? For histology you mean you remove the node and analyze it completely? I recommend to improve the histological results and explain them clearly because cytologies are not always conclusive and this is one of your limitations.
- You have 34 patients and 88 lymphonodes. Knowing that the diagnosis is based on the patient, did you find any discordance between samples from the same cat? This is a really interesting point regarding the repeatability of the process. Did you find different image results in the same animal? Different histology results too?
- I recommend to include a table for the SE results (line 184) instead listing them. Same for PCA results (line 212), please include numbers to support the figure 5.
- A general comment: you compared the B-mode + Doppler results and the Elastography with histology separately. Why you didn't combine everything and get a general score? Maybe putting everything together will give you more precision and give you more specificity to the technique.
There are some spelling errors in the text (like line 232).
Author Response
We thank you very much for your comments,
As follows you’ll find a reply point by point to your comments; the changes in the text are highlighted with the word track changes.
- In Material and methods, line 124, please clarify the criteria for homogeneous/heterogeneous classification of the lymphonodes. A description of the ultrasonographic heterogenicity would be great.
We considered heterogeneous all the lymph nodes characterized by a non-uniform echogenicity of parenchyma as described in Belotta et al. Sonography and sonoelastography in the detection of malignancy in superficial lymph nodes of dogs. J Vet Intern Med. 2019;33(3):1403–13 and Kanagaraju V et al, Utility of Ultrasound Elastography to Differentiate Benign from Malignant Cervical Lymph Nodes. J Med Ultrasound. 2019 Dec 10;28(2):92–8. This aspect has been clarified in the text
- Same section, line 125 and 129, the presence of the Doppler flow can be easily modified with the equipment settings (like PRF's, wall filters, doppler gain, etc...). How can you ensure the total absence of doppler signals? Please clarify the settings you used during the Doppler imaging.
Doppler gain and PRF were properly adjusted to increase the smaller flow sensitivity, this aspect has been clarified in the text.
- For the histological analysis, why they are not all histologies but some cytologies? For histology you mean you remove the node and analyze it completely? I recommend to improve the histological results and explain them clearly because cytologies are not always conclusive and this is one of your limitations.
This information have been included in the text.
- You have 34 patients and 88 lymphonodes. Knowing that the diagnosis is based on the patient, did you find any discordance between samples from the same cat? This is a really interesting point regarding the repeatability of the process. Did you find different image results in the same animal? Different histology results too?
We didn’t find substantial discordance between the samples from the same patients, especially for histopathology results. In some of the patients included in group A we observed different EP from 1 to 3, but since there were no differences in histopathology results and none of the lymph nodes from the same patients was characterized by a predominantly stiff pattern we didn’t focus on this aspect.
- I recommend to include a table for the SE results (line 184) instead listing them. Same for PCA results (line 212), please include numbers to support the figure 5.
SE elastography results have been listed in table 2. Regarding PCA results, providing exact numbers is not the aim of this test, since it provides a graph showing how much the data are different and divided in groups. A clearer explanation of the figure has been included in the text.
- A general comment: you compared the B-mode + Doppler results and the Elastography with histology separately. Why you didn't combine everything and get a general score? Maybe putting everything together will give you more precision and give you more specificity to the technique.
In this work we decided not to include histology in a general score because histopathology was considered the “control test”, the reference necessary to compare the sonographic results to the real anatomy of the lymph nodes.

Round 2
Reviewer 2 Report
I thank the authors for having made the requested changes. However, the text lacks linguistic homogeneity. Often the same concept is indicated with different terms, and this makes it difficult to read and understand the manuscript.
The authors should use a homogeneous terminology,
Eg: in the text the ultrasound score given by B-mode and color-Doppler is sometimes referred to as the "US score" at other times the "B-mode score". I think it is more correct to always call it “US score” or "B-mode and color Doppler score".
Eg: groups A, B and C are sometimes named groups 1, 2, and 3
Line 11: in this line the acronym SE means sonoelastography, in line 23 the same acronym means strain elastography. Authors should use different acronyms to refer to different terms.
Line 11: please replace “with sonography and sonoelastography “ with “by combining B-mode, color Doppler and sonoelastography”
Line 16: please replace “both ultrasound and SE” with “B-mode, color Doppler, and SE”
Lines 17-19: group 1, group 2 and group3 should be renamed “group A, group B and group C, respectively.
Line 20: please replace “B-mode based score” with “B-mode and color-Doppler based score with sonoelastography score”.
Line 20: please replace “sonography and SE” with “B-mode and color-Doppler with SE”
Lines 31-33: groups 1, 2 and 3 should be renamed A, B and C, respectively.
Line 34: please replace “B-mode based score” with “B-mode and color-Doppler based score with sonoelastography score”.
Line 35: please replace “combining B-mode and SE” with “ combining B-mode, color Doppler, and SE”
Line 135: please replace “For analyzing the diagnostic performance of B-mode” with “For analyzing the diagnostic performance of B-mode and color Doppler”
Lines 138-139: please replace “The score for each LN was determined by adding the individual scores. “ with “The US score for each LN was determined by adding the individual scores obtained by B-mode and color Doppler”.
Line 138: please replace “The score” with “The US score”
Line 191: The authors should use a homogeneous terminology, in the text the ultrasound score given by B-mode and color-Doppler is sometimes referred to as the "US score" at other times the "B-mode score". I think it is more correct to always call it “US score” or "B-mode and color Doppler score". Please replace “B-mode score” with “US score”.
Table 2: the table should be followed by a caption explaining the meaning of the acronyms used in the table. In the second to last row of the table, "EP Median (IQR)" should replace "Median (IQR)". The heading B-mode should be replaced by “B-mode and color Doppler” or “Ultrasound” as it contains not only B-mode features but also color Doppler variables (vascularization). For the same reason, the "B-mode score Median (IQR)" should be renamed "US score, Median (IQR).
Lines 289-293: this comment should stay in the main text. Eg. in line 273.
Line 310: Please replace “US” with “it” to avoid repetition.
Lines 357-359: do the authors refer to the sum of the scores obtained by B-mode and color Doppler (US score)? Please specify.
Lines 386-387: are the authors referring to “B-mode score” or “US score”?
Line 392: please replace “a score lower than 6” with “a US score lower than 6”
Line 413: “group 1” should be renamed “group A”
Minor editing of English language required
Author Response
Thank you again for your comments.
As follows, you’ll find a point-by-point response to the changes required
Line 11: in this line the acronym SE means sonoelastography, in line 23 the same acronym means strain elastography. Authors should use different acronyms to refer to different terms.
The acronym SE has benn used only for Strain Elastography and the text has been modified accordingly
Line 11: please replace “with sonography and sonoelastography “ with “by combining B-mode, color Doppler and sonoelastography”
Text has been updated accordingly
Line 16: please replace “both ultrasound and SE” with “B-mode, color Doppler, and SE”
Text has been updated accordingly
Lines 17-19: group 1, group 2 and group3 should be renamed “group A, group B and group C, respectively.
Groups have been renamed in the text
Line 20: please replace “B-mode based score” with “B-mode and color-Doppler based score with sonoelastography score”.
Text has been updated accordingly
Lines 31-33: groups 1, 2 and 3 should be renamed A, B and C, respectively.
Groups have been renamed
Line 34: please replace “B-mode based score” with “B-mode and color-Doppler based score with sonoelastography score”.
Text has been updated accordingly
Line 35: please replace “combining B-mode and SE” with “ combining B-mode, color Doppler, and SE”
Text has been updated accordingly
Line 135: please replace “For analyzing the diagnostic performance of B-mode” with “For analyzing the diagnostic performance of B-mode and color Doppler”
Text has been updated accordingly
Lines 138-139: please replace “The score for each LN was determined by adding the individual scores. “ with “The US score for each LN was determined by adding the individual scores obtained by B-mode and color Doppler”.
Text has been updated accordingly
Line 138: please replace “The score” with “The US score”
Text has been updated accordingly
Line 191: The authors should use a homogeneous terminology, in the text the ultrasound score given by B-mode and color-Doppler is sometimes referred to as the "US score" at other times the "B-mode score". I think it is more correct to always call it “US score” or "B-mode and color Doppler score". Please replace “B-mode score” with “US score”.
We provided a more uniform terminology and we replaced B-mode score with US score comprehensive of both B-mode and Color doppler findings; we specified it in materials and methods and then used this terminology in the text.
Table 2: the table should be followed by a caption explaining the meaning of the acronyms used in the table. In the second to last row of the table, "EP Median (IQR)" should replace "Median (IQR)". The heading B-mode should be replaced by “B-mode and color Doppler” or “Ultrasound” as it contains not only B-mode features but also color Doppler variables (vascularization). For the same reason, the "B-mode score Median (IQR)" should be renamed "US score, Median (IQR).
A caption has been added to the table and the text has been modified as requested
Lines 289-293: this comment should stay in the main text. Eg. in line 273.
The comment has been removed from the text
Line 310: Please replace “US” with “it” to avoid repetition.
Text has been modified accordingly
Lines 357-359: do the authors refer to the sum of the scores obtained by B-mode and color Doppler (US score)? Please specify.
Text has been modified
Lines 386-387: are the authors referring to “B-mode score” or “US score”?
Text has been modified
Line 392: please replace “a score lower than 6” with “a US score lower than 6”
Text has been modified
Line 413: “group 1” should be renamed “group A”
Text has been modified